# Hyper-Activity Books and Serious Games: How to Promote Experiential Learning beyond Distance

**DOI:** 10.3390/ijerph191711132

**Published:** 2022-09-05

**Authors:** Michela Ponticorvo, Elena Dell’Aquila, Raffaele Di Fuccio

**Affiliations:** 1Natural and Artificial Cognition Laboratory Orazio Miglino, Department of Humanistic Studies, University of Naples Federico II, 80133 Naples, Italy; 2Department of Humanities, Literature, Cultural Heritage, Education Sciences, University of Foggia, 71122 Foggia, Italy

**Keywords:** technology enhanced learning, learning materials, serious games, hybrid materials

## Abstract

The COVID-19 pandemic has accelerated the adoption of digital tools for learning. Experiential learning, a crucial process in the educational pathway, can also be promoted at a distance. This paper describes HAB, hyper-activity books and serious games (SG) as methodologies to be used for assessment and training that overcome physical distancing due to COVID-19 in learning. We introduce some examples of these methodologies. The experiences and results represent a pool of resources for experiential learning in everyday educational practice and not merely for responding to emergencies caused by pandemics.

## 1. Introduction

The COVID-19 pandemic has been a significant challenge to education systems worldwide, leading to an inevitable surge in the use of digital technologies. Although the trend of using digital tools in learning was already in place [1], we have witnessed an incredible acceleration in the use of digital technology in the latter two years. This fast introduction has not guaranteed a reflection on how to harmonize digital tools and educational approaches [2]. Educational practices and teaching strategies enabling the involvement of learners, creation of new knowledge and meaningful exchange, and sharing of experiences have not received much attention during the sanitary emergency. Thus, the teaching-learning process radically changed, mostly becoming solitary experiences.

The sudden interruption of classroom learning has affected relationships between peers, the teachers and students, the family and the school, and the learning process with the need to devise alternative approaches to the well-established face-to-face education. This paper does not have the ambition to face all the possible methods that could be applied to address distance learning but to propose not improvised technical and theoretical solutions that can be used to engage students in the online experiential learning process.

## 2. Theoretical Reference

To describe what has been ignored during the COVID pandemic regarding psycho-pedagogical theories, collaborative learning, peer education, and experiential learning must be cited. Even if this paper is not meant to discuss in detail these theoretical frameworks, it can be useful to introduce the key elements of these approaches.

Collaborative learning [3] and peer education [4] were put aside. Collaborative learning theory entails peer-to-peer learning that stimulates deeper thinking among learners.

Peer-to-peer educational experiences help fill the students’ gaps in storing and delivering insights into subject matter to gather the aspects that they did not grasp previously [5]. They can join the discussion more responsibly with their peers to jointly organize and develop more information related to specific topics and propose new ways to address it to the group members.

Moreover, according to the constructionist approach [6], learners learn best by making tangible objects within authentic learning opportunities that allow for a guided, collaborative process that integrates peer feedback.

The constructionist approach [7] emphasizes learners’ agency as active participants in constructing their learning rather than just taking in information passively. This approach argues that learning occurs more effectively when learners are involved in producing tangible and shareable objects. By experiencing the world and reflecting upon those experiences, learners build their mental models to understand the world around them and integrate new information into their pre-existing knowledge.

Experiential learning [8,9] emphasizes the central role of experience in every learning process and provides hands-on experiences throughout lifelong learning.

Proposing activities that enable a student to learn while engaged in solving a real-life problem or practicing in a safe context, for example with games is a good implementation of the “learning by doing” principle [10,11,12].

COVID-19 has posed the challenge to keep the pros of these educational approaches thanks to digital technologies in every learning/teaching context (school, university, adult training).

In this paper, we aim to introduce two possible methodologies that can be used to face this challenge, as they are based on at-distance experiential learning activities: hyper-activity book (HAB) and serious game (SG).

## 3. HAB: Hyper-Activity Book for Children

### 3.1. The School Context

The diffusion of new technologies has redefined the borders within which the experience of learning happens in terms of space and time. The increased accessibility to high-quality occasions and materials has led to a paradigm shift in education, such as the flipped classroom [13] embraced with conviction, especially in adult education. In the school context, especially in the first grades of education, the digital revolution has not fully expressed its potential for the specificity of the processes and mechanisms of age group learning that we identify with in childhood [14]. The application of new technologies in school has found a few obstacles. These are related to the effective applicability of technologies in class contexts, either for the peculiarities, opportunities, limits, and available resources related to this scope. Thus, this requires a reflection to integrate hints from the psychology of development, educational psychology, and pedagogical practices [15].

The Coronavirus pandemic has raised the awareness that, if driven by necessity, kindergarten and primary school can exploit e-learning for daily didactic activities [16,17]. In the past two years, a huge number of children, teachers, and parents could experience what researchers have argued for decades [18]: to educate, train, and teach at a distance is not enough to stand in front of a camera and give the lesson as if you were in a normal classroom [19].

Psychology and pedagogy advise that the younger learners learn with constant and playful exercise based on manipulation of physical objects, multisensory stimulation, imaginative narration, on cooperative social play [20]. It is not possible to propose this kind of experience through a lesson (either asynchronous or synchronous) adopting traditional e-learning methodologies [21].

During the COVID-19 emergency, teachers and other school professionals have deservedly activated distance learning supported by various e-learning platforms in a very short time. Still, they drastically resized experiential learning [22], playing a central role in the learning process.

The inadequacy of current teaching tools can explain the difficulty of teaching at distance in kindergartens and primary schools. Exercise books, notebooks for homework, educational games, multisensory teaching materials, and structured materials, for example, Montessori ones, are not fit to be used at distance [23].

Indeed, well-established psycho-pedagogical approaches, such as the Montessori one we have cited, have proposed some articulate didactic methodologies focusing on children’s active involvement and supported by learning materials, which foster both learning and teaching processes by stimulating concrete objects manipulation and peer group cooperation: as a result, such approaches foster an active education practice that is particularly useful for the acquisition of cognitive and social skills [24,25].

One relevant element of such psycho-pedagogical methodologies is the chance to interact with physical educational games, such as logic blocks, cards, teaching tiles, abacus, physical representations of letters and numbers, etc., aimed at training specific perceptive, cognitive, and motor functions [26]. As highlighted earlier, these materials are conceived to be used in the presence and they require a re-thinking to be used at distance.

For this reason, it is necessary to enhance existing experiential teaching tools and make them usable remotely, availing of the support from teachers. A possible way to do this is represented by the HAB methodology, which we will introduce in the next sub-section.

### 3.2. What a HAB Is

In a worldwide primary school/formal learning context, widespread learning materials are grounded on the principles of valuing interactions between teachers/learners and favoring active pupils’ participation and involvement. These learning materials are mainly: (1) schoolbooks to be used with paper, pens, and brushes and (2) educational materials that come from the learning by doing and Montessori tradition [27]. They are indeed effective and well accepted by teachers and children, however, it is not possible to use them in every condition and, above all, it is not easy to personalize activities for each child. This is particularly true and pervasive in disadvantaged contexts and with children with special needs.

Some technologies that will be detailed later can be used to upgrade and innovate exercise books to make them become what we call hyper-activity books or HAB [28,29,30].

The hyper activities book (HAB) is an exercise book that about the hidden layer [31,32] is completely identical to its equivalent on paper; and regarding the external layer can employ paper form or other human-computer interfaces (digital, physical or hybrid).

HAB can be remotely controlled by teachers both artificial and humans [32]. In schools, teachers, educators, and lecturers often propose activities to perform in (relative) autonomy. Typically, such activities are based on the use of exercise books, which can be considered, in this context, fundamental elements of the student-teacher relationship. The starting point is represented by the teacher who selects from a workbook some activities (tasks) to be carried out with autonomy by students. Hopefully, the assignment of tasks to each learner should be personalized. The results and reactions of the students are observed and analyzed by the teacher, who uses this information to plan subsequent activities, including assigning new tasks [29,32].

Parents and other people who support the children in the educational pathway are informed on results and can monitor the educational pathway.

HAB architecture is represented in Figure 1.

Figure 1 shows some relevant HAB features:It integrates multimedia technologies and multimodality interaction with natural interfaces, smart objects, and environments;It includes intelligent services, namely adaptive tutoring systems and learning analytics tools;It exploits the integration of digital technologies and contents with physical objects and environments.

HAB is a model that allows application with different technologies. It well represents what is described as a hybrid environment [33] that combines digital and tangible interfaces. HAB runs inside the digital learning environment, choosing the right exercises, and orchestrating the management of single exercises.

On the interface, the system provides multimedia, a file, or content outputted by the digital interfaces with specific feedback. It could be typically aural or visual, delivering to the user/s the information for completing the task. The user interacts with the hyper-activity book with a tangible interface. It can be a simple keyboard as in the case of virtual reality environments or touchscreen technologies when mobile devices are used.

Nevertheless, some other interesting approaches are adaptable to the HAB model. Tangible user interfaces–TUIs [34]—are smart objects empowered with specific sensors that allow quick recognition by digital interfaces. The TUI’s strength is that the user interacts with real and physical objects in the environment. By manipulating these objects with hands, the user drives the digital interfaces to perform the task assignment as the system requires. Tangibles give physical form to digital information [34], representing and controlling their digital counterparts. This way the user interacts with a physical space with all senses in line with the Montessori pedagogy, previously cited. If the tangible interface is an object that elicits the olfactory sense, the user needs to perform a multisensory approach during the learning [35].

HAB is not limited to the Tangible User Interface paradigm, as users can interact also with handwriting recognition, operated by digital interfaces Numerous tools enable handwriting on digital media, e.g., touchscreens, graphic tablets, and pens [36].

As shown in Figure 1, the HAB model includes another element that has an interface with the outer part: it produces reports. Reports and feedback on the learning results and students’ behavior are released with learning analytic modules [37]. HAB conceptualizes learner data tracking by recording each activity during the session. This way, the teacher can access the personal control panel to check the raw tracking data collected from the platform or compare the learner data with previous sessions, class results, and the learning objectives set by the teacher.

Each activity is intended to stimulate a particular knowledge domain: for instance, there are exercises on math and logic and activities to develop imagination and creativity. Teachers can select and plan activities through a dedicated, easy-to-use- interface, or this function can be run automatically by tutor 0 (see Figure 1).

Once the exercise/activity is chosen, planned, and properly configured and adapted, the child can begin. It is worth emphasizing that the activities are carried out in cyber/physical environments, merging, and integrating digital contents and tools with physical (smart or not) objects, thus enriching digital environments with motivational and emotional factors to enhance learning effectiveness. The planned narrative or story will attract the child and immerse him/her in a completely different environment: this aspect is relevant in every educational context [38]. From a cognitive psychology perspective, the narrative level exercises a framing effect on another level corresponding to the problem space, i.e., the engine [29].

The narratives provide different perspectives, which change problem perception and the activity itself. Depending on the type of activity, the engine can be conceptualized as a concrete Euclidean space (chess or checkers game) or as a logical, non-geometric structure. In both cases, the engine structure is configured as an interaction between the player (or the players in a collective scenario) and the problem space that modifies the problem state. In the activity layer, while playing the child interacts with physical materials, Montessori-like educational materials, and the tutor, which aims to maintain a high interaction level from the child. This is achieved through a particular kind of interaction that is multimodal interaction, mediated mainly by tutor 1 in Figure 1.

Multimodality is mainly at the service of the narrative level: relying on proper graphical solutions, it keeps the child in the fiction dimension, for example with a mascot character introducing and commenting on activities. Multimodality leads to natural interfaces that are invisible to the user and allows continuous interactions without using artificial control devices whose operation must be acquired.

The tutor named T2 (see the figure above) builds an individualized report on child interaction (invisible to the child user), records achievements and failures, and activity preferences to produce a detailed user profile. This report returns to the teacher who can further customize HAB/child interaction with a focused choice from the database and to T0 that can better adapt itself to the child. In this way, the cycle can start again.

It is evident that the role of tutors is fundamental to the model: they implement the so-called learning analytics that measure, collect, analyze and report data about learning in the activity layer to support learning processes. In HAB the tutors are artificial agents that exploit their artificial intelligence to mediate learning analytics.

In summary, the tutors unfold different functions in interaction with the learner. They can select and propose ad hoc exercises and activities fitting with different learners’ needs, thus going beyond the standardization of educational materials. Teachers can create personalized educational resources and plan/adapt learning activities (e.g., adding video to explain concepts and vocal comments, creating new stories, redefining learning objectives, etc.). Furthermore, they can develop new material (cards recognizable by the tablet by uploading their images, daily life objects to start a learning activity, etc.) for the different emerging learning needs.

Moreover, teachers can use and offer tools to examine the learner’s performance. Indeed, adaptive tutoring systems and learning analytics tools give the learner immediate and appropriate feedback based on individual learning patterns. Learning analytics work at the single child level, proposing data analysis about the last session and the previous exercises, and at the group level by aggregating data collected and related to one or more specific groups. These analytic tools will also suggest some parameter changes that the teacher can decide to apply. Moreover, a report is delivered with different aggregation levels (class, school, population, etc.) to assess a global evolution of a given group.

Integrating digital technologies and contents with physical objects and environments is crucial as it relies on action and related sensory and motor achievements. Motor actions, exploration, and experimentation match and support their learning processes, which, along the growth process, are gradually simulated in the human mind and become symbolic and cognitive acts. Despite this, “action” covers a very relevant role in every human life phase. The Embodied and Situated Cognition Theory, ESCT [39,40,41] has underlined this aspect in recent years. Action is indeed a need for human knowledge.

HABs combine digital and physical elements with being used in experiential learning even when the teacher is physically distant from the learner. This is thanks to the integration of teaching/learning models implemented in technology. They are therefore a suitable vehicle for promoting remote experiential teaching an example of HAB implementation.

### 3.3. Block Magic: A HAB Example

Block Magic (Figure 2) is an example of HAB that derives from the Dienes’ blocks [42]. It consists of a set of magic blocks (48 traditional logic blocks), a magic board/tablet device, and specific software, based on STELT [43]. The technology used in Block Magic is the RFID/NFC, Radio Frequency Identification/Near Field Communication. An RFID system consists of an antenna and a transceiver, which can read the radio frequency and transfer the information to a device, and a small and low-cost tag, which is an integrated circuit containing the RF circuitry and data to be transmitted. The blocks are made up of a set of blocks (usually 48 pieces) divided into four groups according to different attributes: geometric shape (triangular, squared, rectangular, and circular), thickness (thick and thin), color (red, yellow, and blue) and dimension (big and small). Traditional logic blocks are equipped with RFID tags. This configuration permits to a PC or a tablet, with BM software installed to connect with BM Magic Table, another relevant BM material. The Magic Table has a hidden antenna that recognizes each block, sends a signal to the PC/tablets, and produces feedback coherently with pupils learning path. Each augmented magic block had an integrated/attached passive RFID sensor for wireless identification of every single block. A specially designed wireless RFID reader device, an active board, is used, which could read the RFID of a block and transmit the result to the BM software engine. Activities are on logic, mathematics, languages, etc. The two parts mainly form the BM software engine first one is devoted to receiving input from the active board and generating an “action” (aural and visual). These actions implement the direct feedbacks the user can receive interacting with the system. These feedbacks are regulated by an Adaptive Tutor System embedded that ensures autonomous interaction between the user and the system, receiving active support, corrective indications, feedback, and positive reinforcement from the digital assistant on the outcome of the actions performed. Adapting tutoring systems [44] is an Artificial Intelligence application that provides instruction tailored to individual learners’ needs. The second software component is also devoted to customization. It is dedicated to teachers, educators, etc., allowing them to choose the exercises to be proposed to the child, focusing the attention on the skills the child needs to train more. Moreover, the BM software can collect data about the exercises.

## 4. Serious Games

### 4.1. The Context

SG is receiving increasing recognition in training and education for its capacity to provide experiential learning opportunities. Such trainee-centered activities can efficiently facilitate cooperative learning and strengthen students’ participation, connection and understanding of learning material [45]. E-learning systems can provide the scaffolding for designing educational SG to promote soft, transferrable skills development in the school context. The development of transferable skills favors students to become active learners and citizens capable of facing personal, academic, and social competencies in contemporary challenges and future trends. Moreover, the acquisition of transferable skills is important for the development and reinforcement of other skills.

For SG, we intend games specifically designed for learning purposes, so called EduTechRPG (Technologically Enhanced Educational Role-Playing Game) to develop soft skills, considered essential and complementary to subject-related competencies [31] enabling people to be flexible and adaptable in different roles or in different situations inspired to real life. Soft skills refer to relational competences expressing personal ways to manage and approach relationships with others and our reactions. EduTechRPG represent the migration and adaptation of the traditional psycho-pedagogical methodology, such as role-playing and psychodrama [46] to digital and online environments. EduTechRPG allows users to experience direct involvement with the learning objectives. They can act out roles, competences and use various communication technologies allowing for a virtual extension of the action space: the scene of traditional psycho-pedagogic role-playing is extended to the stage to virtual environments. It is widely recognized that to effectively learn and develop transferable skills, there is a need for a real-life context to learning that cannot be solely experienced in teacher-led or classroom-based activities. Instead, teachers can act as facilitators and enhance SG-based learning by playing with students in virtual environments in which meaningful experiences are available [47].

### 4.2. Role-Playing Serious Games

It is generally acknowledged that role play is particularly suited to experiential learning. This powerful tool enables participants to draw into an experience and move the learning experience from an impersonal, theoretical, and notional form into interactive and participative dimensions. The Role-play technique goes beyond experiential learning. As it uses the art form of dramatization on an educational stage, it boosts learners’ awareness of self and others, enhances mental flexibility, and creates multiple perspectives extending thoughts and feelings beyond the horizon of personal interpretation.

EduTechRPG represents an innovative learning methodology for soft skills training through the migration and adaptation of the psycho-pedagogical methodology, such as role-playing and psychodrama, to digital and online environments.

In literature, it is possible to find diverse examples of how role-play methodology can be transferred to virtual environments. Some have been used to complement face-to-face activities or may provide stand-alone solutions and conceive single-player or multiplayer settings available [32]. These are generally characterized by conversational and emotional interactions or interactions mediated by objects and action exchange. Depending on the underpinning learning approach, they can involve the achievement of learning objectives through a personal dramatization, or goals that require performing a certain number of actions to be achieved [31].

Thus, these tools allow different forms of active methods that can be flexibly employed for different contexts of application (professional, vocational, and educational contexts) to develop a variety of skills (procedural knowledge, technical and vocational skills, soft skills).

EduTech RPG are flexible artifacts in the hands of teachers and students for reproducing and experiencing real-world context enabling the development of skills for learning, life, and work.

The methodological approach is characterized by a set of pedagogical elements that represent its specificity in terms of a meaningful learning environment, such as Intelligent tutorship, psychological modeling, and feedback mechanisms.

They represent an innovative form of active learning that provides users with unique open-source e-learning tools for assessing and training transversal competencies to which they can benefit, overcoming the common lack of access to affordable training and developmental resources.

In the actual society and digital era, these systems can provide the scaffolding to transfer, develop and implement innovative practices relying on EduTechRPG to enhance key competences (e.g., cultural awareness and digital creativity) of key actors in the education sector (teachers, students, educators).

The EduTechRPG we present in this paper relies on a methodological approach based on the use of learning environments and resources, i.e., MOOC and virtual scenarios, ACCORD, tackling interpersonal conflict management and effective communication in intercultural and interethnic educational settings. Such innovative methodology builds on previous EU projects, such as ACCORD and has been proven as useful and effective virtual role-play to transfer soft skills in the school context used by teachers available [48].

The platform allows teachers to play virtual scenarios for assessing personal and students’, intercultural and interethnic competencies and to create personalized learning environments according to specific learning needs. The strength of such virtual learning resources resides in its flexibility as they can be run and employed either online or face-to-face in the classroom.

Under the guidance of teachers, the platform enables different groups of students to play with the same scenarios to share diverse contexts and situations experienced and promote respect for the diversity of communication styles, diversity related to ideas and cultural expressions, ownership of shared values, equality, and non-discrimination.

In terms of theoretical modelization and operationalization ACCORD game is based on the theory of interpersonal conflict management proposed by Rahim and Bonoma [49]. ACCORD is a 3D serious game to train and assess users’ negotiation and interethnic communication skills in realistic scenarios during interactions with artificial agents (BOT). It provides an Intelligent Tutoring System that uses the data collected during the interaction to generate tailored feedback guiding users to improve their skills [50,51]. Moreover, the MOOC ensures the availability of both content and digital resources related to the competencies targeted to be developed, including virtual learning scenarios, video lessons, and multimedia resources.

In the context of this paper, we aim specifically to explore the application of the ACCORD EduTechRPG in the school context through the diffusion of innovative approaches to make education fit for the digital age and aiming to address the development of cultural awareness and creativity, competencies considered central to answer to the changing needs also due to the actual migration and multicultural configurations of the modern EU society.

By focusing on the use of remote ICT, SG answers to the needs of the actual educational and professional contexts needing remote solutions. This seems particularly true in a moment in which the SARS-CoV-2 virus pushes all the professional world to turn to online resources.

### 4.3. ACCORD: A SG Example

ACCORD is a single-player 3D role-play game intelligence-based tool to train users on interethnic conflict management realistic scenarios during the interaction with artificial agents. The game focuses on the simulation of verbal interaction between two characters (a virtual agent controlled by a human player, and a BOT computer-controlled interlocutor), in which behavioral characteristics such as the act of speech and some elements of body language play a fundamental role [52].

The users can play ten different conflicting scenarios with seven different characters equally grouped per gender and different ethnic variables.

The ACCORD theoretical approach and its simulation game scenarios are based on the conceptualizations of Rahim [53] combined with aspects of the assertive model of communication [54], the theory of universal basic emotions [55,56], and the Multicultural Personality theory [57]. These models have enabled the design of virtual agents’ emotional, physical, behavioral, and psychological characteristics respecting principles and variables derived from those models. A detailed description of the design of the SG-ACCORD Virtual Agents can be found in Dell’Aquila and colleagues [58].

With regards to Rahim’s model, it describes five possible styles of handling interpersonal conflicts resulting from the combination of two basic dimensions: concern for self (degree (high or low) to which a person attempts to satisfy his or her own concern) and concern for others (degree (high or low) to which a person attempts to satisfy the concern of others:Integrating style (high concern for self and others) involves collaboration between teachers and students that can think about available resources to use in creative solutions to reach a mutual constructive and acceptable solutions.Obliging style (low concern for self and high concern for others), also known as accommodating style, is associated with the attempting to emphasizing commonalities to satisfy the concern of the other party that may result useful when teachers are unconcerned about a specific outcome and more interested in preserving classroom relationships.Dominating style (high concern for self and low concern for others) although it is identified with the orientation to win one’s position this style may be appropriate choice when an immediate urgent action o decision is needed based on teachers’ knowledge, skills, or experience.Compromising style (intermediate in concern for self and others) involves both parties give up something to reach a mutually acceptable solution.Avoiding (low concern for self and others) as this style has been associated with withdrawal or sidestepping situations can hardly be considered as an appropriate style of intercultural conflict management to be used by teachers.

The ACCORD virtual scenarios simulate interethnic conflicts between teacher and students within realistic school context scenarios that can be effectively used to engage students in classroom activities synchronously and asynchronously.

The exchange between BOT and user’s AVATAR is organized in one turn five/seven state scene, during which the user can choose one among five/seven possible sentences (one for each of Rahim’s styles of handling conflicts: Integrating, Compromising, Avoiding, Obliging, Dominating, Obliging Appropriate, Dominating Appropriate) that are complemented with behavioral characteristics.

The innovative aspect of the game is represented by the assessment tool expressed by the tutoring system that is available after each game scenario. It provides overall feedback regarding the user’s performance while interacting with the BOT and managing the specific conflicting situations in the game scenario. The user is given a profile based on the Rahim model with specific feedback on the communication style and the efficacy of the solution achieved (Figure 3). The profile emerges through a comparison of the behavior of the user and the style of the artificial agent she interacted with. Moreover, the user is also provided with a history of all the exchanges during the interaction and guided by the tutoring system through the understanding of the effectiveness of the choices made.

The intelligent tutoring systems provide guidance and feedback on the user’s performance, enabling reflection and self-awareness and thus ensuring transference of learning.

The user (teacher or student) can decide to play the same scenario for as long as she wants, reflect on the choices made, and engage in the most effective conflict management and communication results.

Moreover, teachers can assist learners while playing the scenarios to provide additional feedback and analyze the user’s entire performance considering the specific learning objectives.

Therefore, consideration of designing SG for learning purposes to be introduced in school contexts instead of using commercial-off-the-shelf role-play games resides in their effective support in learning [59], as well as skill assessment [60].

## 5. Conclusions

Our complex and globalized society requires us to conceive the opportunity to support experiential learning even beyond the (physical) classrooms.

In this paper, we have tried to delineate some possible pathways that can be followed to overcome the physical distance between teachers and learners and among learners. This issue, indubitably evident during the pandemic, can affect learning processes in different contexts. The aim of this work was neither to report the results of the trials we have conducted in various EU schools nor to show the methodology we have designed to test the effectiveness of the described digital tools. These results can be found in previous works [35,42,48,50,51,52]. Our main aim was to contribute to the debate on how existing and future tools can be employed in physical distance education.

The need to overcome the distance was already necessary for children living in geographically isolated areas or unable to attend school for various reasons (illnesses, parents on long business trips, etc.). As technology is at the core of our lives, it is essential that we leverage it to provide engaging and powerful tools and resources that make meaningful and authentic learning experiences for students. Virtual connections between people and technology resources and tools can reduce time and distance barriers in the physical world. Thus, it urges our educational system to empower the use of the existing tools specifically developed for experiential e-learning activities, as they become tools for changes in the teachers’ hands.

In future research, we will explore how HABs and SGs affect and change learning processes, which conditions allow them to exploit their potentialities better, and which variables must be considered to maximize impact at different ages.

## Figures and Tables

**Figure 1 ijerph-19-11132-f001:**
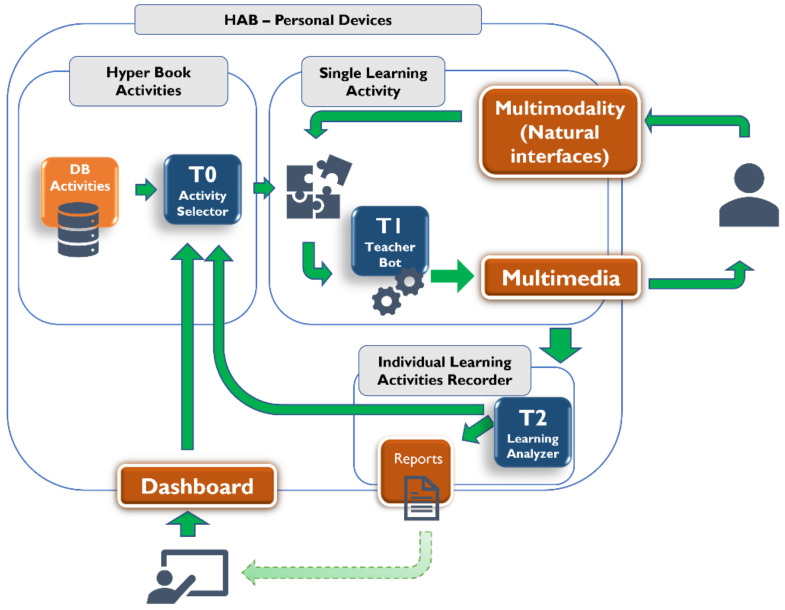
HAB architecture.

**Figure 2 ijerph-19-11132-f002:**
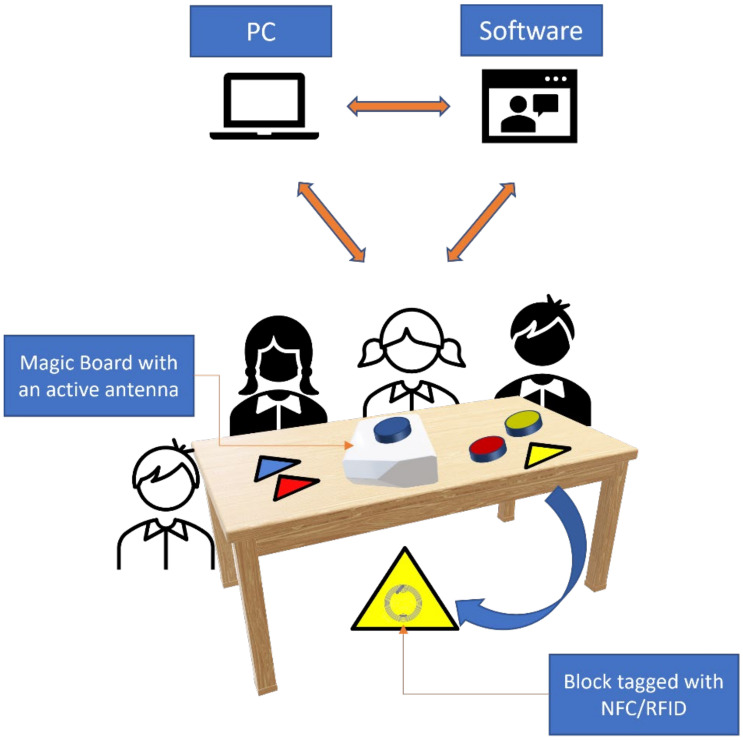
Block Magic architecture.

**Figure 3 ijerph-19-11132-f003:**
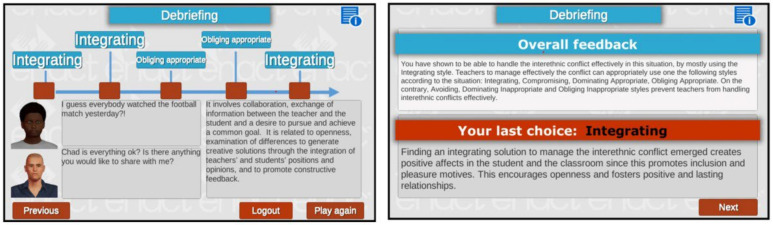
ACCORD debriefing system.

## Data Availability

Not applicable.

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
