# Peer review of "Hyper-Activity Books and Serious Games: How to Promote Experiential Learning beyond Distance"

_ijerph, 2022, doi:10.3390/ijerph191711132_

Round 1

Reviewer 1 Report

The paper is interesting, suggestive and well written. The arguments are coherent, the description of the devices are correct, the theoretical background is solid. The problem is that I do not appreciate an empirical research on the topics, only a description of the devices, but the devices are not tested in an empirical research with users, so the claims are exposed at a theoretical level. 

Author Response

Thank you for your time and effort in reviewing our paper. We appreciate that you found some merit in it. We agree that the claims are exposed at theoretical level, as we meant to describe the approach. Nonetheless, in previous papers we have inserted as reference, there are descriptions of the empirical tests with users.

Reviewer 2 Report

The challenge of experiential learning in online environments is a very relevant to our context. Some good points are made.

To strengthen the paper please consider the following:

- Use the Introduction to set the context and make clear the research question or problem that is being discussed. There is some mention of various learning theories in the introduction. Instead of attempting to cover the learning theories briefly, in point form just in the introduction, consider adding a theoretical frame section where experiential learning is discussed and the premises of experiential learning are more clearly and more rigorously explained. This is somewhat done in the school context but I feel it would add clarity to specifically discuss underpinning theory.  Also clearly state the research question or problem statement in the introduction i.e. how we can engage students in experiential learning online.  

- there are a lot of concepts throughout the paper other than experiential learning e.g. tangible interfaces, diffusion of innovation. Consider how the paper can be restructured to create a smoother flow or argument 

- At the conclusion it is stated "In this paper we have tried to delineate some possible pathways that can be followed to overcome the distance between teachers and learners and among learners". - The goal could clearly be stated in the introduction to provide a clearer structure 

- It is good to explore the possibilities of different technologies. There are some good points made in the paper. To realise the aim of the paper and strengthen the contribution of the paper, re-work the line of argument to create a smooth flow in the paper. For example, start with the need for exploring technologies to create engagement and experiential learning, discuss theoretical principles or in other words provide a conceptual framework, then undertake a critical review of the technologies (rather than descriptive). A more structured literature review may be useful.  Hope these comments are useful.

Author Response

Thank you for reviewing our paper and giving us useful comments and suggestions: we have tried to apply the suggested changes in the text flow and organization. We have added a new section, widened the introduction to better explain our paper goals and re-written conclusions. We have not introduced a new literature review because, in our opinion, this does not fall within our paper scope, that is to show some pathways to keep experiential learning at distance and not to critically revise all possible solutions.

Round 2

Reviewer 1 Report

Now the paper is more clear regarding the purposes and aims. However, it has to be clarified that the authors are the designers (or not) of the examples explained. 

Funding: BlockMagic was funded by the European Commission under the call LLP-Comenius grant 529 number 517936-LLP-1-2011-1-IT-COMENIUS-CMP. 530 ENACT was funded by the European Commission under the call Erasmus+ KA3 grant number 531 580362-EPP-1-2016-1-IT-EPPKA3-IPI-SCO-IN.

Author Response

Thank you for your feedback. We have now added a note to clarify that authors were involved in the design and implementation of the examples. 
